# Lung Hyaluronasome: Involvement of Low Molecular Weight Ha (Lmw-Ha) in Innate Immunity

**DOI:** 10.3390/biom12050658

**Published:** 2022-04-30

**Authors:** Antony Hoarau, Myriam Polette, Christelle Coraux

**Affiliations:** Inserm UMR-S 1250, University of Reims Champagne-Ardenne (URCA), SFR Cap-Santé, 51100 Reims, France; antony.hoarau@univ-reims.fr (A.H.); myriam.polette@univ-reims.fr (M.P.)

**Keywords:** hyaluronic acid, lung, inflammation, innate immunity

## Abstract

Hyaluronic acid (HA) is a major component of the extracellular matrix. It is synthesized by hyaluronan synthases (HAS) into high-molecular-weight chains (HMW-HA) that exhibit anti-inflammatory and immunomodulatory functions. In damaged, infected, and/or inflamed tissues, HMW-HA are degraded by hyaluronidases (HYAL) or reactive oxygen species (ROS) to give rise to low-molecular-weight HAs (LMW-HAs) that are potent pro-inflammatory molecules. Therefore, the size of HA regulates the balance of anti- or pro-inflammatory functions. The activities of HA depend also on its interactions with hyaladherins. HA synthesis, degradation, and activities through HA/receptors interactions define the hyaluronasome. In this review, a short overview of the role of high and low-molecular-weight HA polymers in the lungs is provided. The involvement of LMW-HA in pulmonary innate immunity via the activation of neutrophils, macrophages, dendritic cells, and epithelial cells is described to highlight LMW-HA as a therapeutic target in inflammatory respiratory diseases. Finally, the possibilities to counter LMW-HA’s deleterious effects in the lungs are discussed.

## 1. Introduction

Hyaluronic acid (HA) was discovered in the Department of Ophthalmology at Columbia University, New York, in 1934 in the bovine vitreous humor [1]. HA is an unsulfated glycosaminoglycan consisting of glucuronic acid and glucosamine disaccharide repeating units linked together by glycosidic bonds. It can reach a weight of about 10^7^ Da and a size of about 25 µm. HA also has important viscoelastic properties: each HA molecule has hydrophilic and hydrophobic endings, and, thanks to the complementary poles, the HA molecules assemble to form a polymer. This characteristic allows the formation of a gel with the viscoelastic properties necessary for effective medical options to treat joint pains [2]. The main specificity of HA is its high affinity for water. HA is able to absorb up to one thousand times its weight in water with an unbranched linear topology. In addition, HA is a critical component of the extracellular matrix (ECM) that regulates the development, normal structural integrity, as well as responses during injury, repair, and regeneration [3]. The activities of HA are mediated through interactions with cell surface receptors called hyaladherins, including Cluster of Differentiation-44 (CD44), Toll-like Receptor 2 and 4 (TLR2, TLR4), Receptor for Hyaluronan-Mediated Motility (RHAMM), Hyaluronic Acid Receptor for Endocytosis (HARE), and Tumor Necrosis Factor-Stimulated Gene-6 (TSG-6). Under physiological conditions, HA is synthesized as High-Molecular-Weight-HAs (HMW-HAs) that assume anti-inflammatory functions, mainly through CD44 interactions [4]. Infectious and inflammatory environments lead to HMW-HA fragmentation into Low-Molecular-Weight-HAs (LMW-HAs) that activate pro-inflammatory signaling pathways. This effect is associated with interactions, primarily with TLR2 and TLR4, and HARE receptors. Their presence in plasma membranes could explain the pro-inflammatory function of LMW-HA. Finally, HA interacts with RHAMM, a microtubule-associated protein involved in cell motility, transformation processes, and cell growth and proliferation [5]. The synthesis, degradation, and activities of HA through HA–receptor interactions define the hyaluronasome that represents a functional unit leading to cell response depending on its environment. The hyaluronasome remains dynamically sensitive to extracellular and intracellular stimuli. Therefore, a specific response related to the size of HA could provide information about the cell environment. These size-dependent roles prompted us to discuss the influence of HMW-HA/LMW-HA imbalance in the lung.

## 2. Physiological Context and HMW-HA

### 2.1. HMW-HA Synthesis

HA is present as HMW-HA at a steady state. It is synthesized by fibroblasts, epithelial and immune cells, as well as mesenchymal and hematopoietic stem cells [6]. Their production is regulated by the transmembrane enzymes hyaluronan synthases (HAS). In humans, three different genes located in the 19q13.4, 8q24.12, and 16q22.1 loci encode the HAS1, HAS2, and HAS3 isoforms, respectively. The enzymatic activity of HAS operates on the inner side of the plasma membrane and extrudes the forming polymer to the extracellular medium [7]. This mode of HA production allows the extension of the disaccharide repeats necessary to obtain HMW-HA. The turnover of HMW-HA is rapid, with a complete renewal in less than a day [8,9]. The molecular weight of HA produced in vitro by HAS3 in fibroblasts is 10^5^–10^6^ Da, while production by HAS1 and HAS2 is at least twice as high [10]. It has been reported that fibroblasts overproduce HA during their proliferation, in comparison to their resting phase. Moreover, cell proliferation is specifically dependent on neosynthesized HA [11]. HA production is regulated by different stimuli, including Platelet-Derived Growth Factor (PDGF), Transforming Growth Factor- β1 (TGF-β1), Tumor Necrosis Factor-α (TNF-α), and Interleukine-1β (IL-1β) [12,13]. In addition to HAS, enzymes such as uridine diphosphate–glucose dehydrogenase (UGDH) and uridine diphosphate–glucose pyrophosphorylase (UDGP) may also promote HMW-HA formation [14].

### 2.2. Beneficial Roles of HMW-HA

The HA properties have been described based on their molecular weight. In their native form (>1000 kDa), HMW-HA interactions with ECM components play an important role in the ECM’s stability and structural organization. They promote cell quiescence and support tissue integrity. Indeed, through interaction with CD44, they lead to cell cycle arrest [15]. Via CD44, HMW-HA enhanced vascular integrity in vitro and in vivo in a murine model of LPS-induced acute lung injury [16]. It protects cells against apoptosis in the intestinal epithelium: irradiated mice are characterized by increased expression of HA synthases and HMW-HA in the intestine and the plasma. The injection of HMW-HA before radiation leads to the survival of the intestinal crypt and a decrease in radiation-induced apoptosis mediated by TLR4 [17]. Moreover, they can decrease both β-hexosaminidase secretion and histamine release, and block degranulation in the basophilic leukemia of rats [18]. Very HMW-HAs (vHMW-HAs) could also exhibit interesting properties in cancer research. They were associated with cancer resistance in a model of a naked mole-rat [19]. HMW-HAs participate in the oral wound healing process by forming a scaffold suitable for periodontal tissue regeneration [20,21]. They have also been extensively studied in aging-associated diseases, such as knee arthritis or rheumatoid arthritis. HMW-HAs participate in the maintenance of the viscosity of the articular liquid, as well as in the maintenance of the integrity of the cartilage via their interactions with aggrecan [22,23]. Through the CD44 receptor, HMW-HAs also induce osteoclast differentiation via the activation of the Rho kinase–NF-κB pathway [20]. HMW-HAs have extensively been described as anti-inflammatory mediators by suppressing cell–cell interactions, or ligand access to cell surface receptors. As an example, HMW-HAs inhibit periodontal inflammation by preventing *P. gingivalis*-induced activation of IκBα, extracellular signal-regulated kinases (ERK1/2), and mitogen-activated protein kinases (p38MAPK) [21]. Moreover, they improve synovial inflammatory pain by suppressing the CD44-mediated IL-6-induction of matrix metalloproteinase (MMP) secretion in human chondrocytes [24]. HMW-HAs exert anti-inflammatory effects on renal, nervous, or ENT (Ear, Nose, and Throat) tissues [25,26,27] and, by downregulating TNF-α, IL-8, and inducible Nitric Oxide Synthase (iNOS) expression, on fibroblast-like synoviocytes [28]. At the respiratory level, HMW-HAs suppress the polarization of M1 macrophages, improve the production of IL-10 in lungs inflamed by PM 2.5 particles and promote decreases in p38 and JNK phosphorylation [29,30]. HMW-HAs lead to the inhibition of prostaglandin E2 (PGE2) macrophage-associated production by suppressing lipopolysaccharide (LPS)-induced Cyclooxygenase 2 (COX2) [31,32]. HMW-HAs ameliorate respiratory failure in acute COPD exacerbation by lowering systemic inflammation markers in a clinical trial [33]. This raises questions about their potential synergistic action associated with other anti-inflammatory drugs. Their biological effects could be at the origin of an interesting therapeutic strategy in the context of inflammatory lung diseases. 

### 2.3. Therapeutic Use of HMW-HA 

HMW-HAs represent good candidates for therapeutic strategies in tissue regeneration, dermatology, or orthopedics. Their rheological properties facilitate cell proliferation, migration, and wound healing [34]. HA serves as a temporary matrix that allows, mainly due to its large molecular size, the diffusion of nutrients and waste products into the site of injury. Several important functions associated with HA during the wound healing process originate from its interaction with the CD44 receptor [35]. HA directly affects the proliferation and migration of keratinocytes [36]. These findings led to the development of HA-based biomaterial (HYAFF^®^) as a scaffold for tissue repair. This biomaterial allows cellular invasion and capillary growth, processes that are fundamental for skin re-epithelialization. It has been indicated for the management of partial- and full-thickness wounds, second-degree burns, pressure ulcers, venous ulcers, chronic vascular ulcers, as well as surgical and trauma wounds [37]. HMW-HAs represent high interest in a gel-like form in chronic inflammatory diseases, such as joint pain or rheumatoid arthritis. This gel acts as a lubricant and shock absorber for the joint and tendon structures [38]. HMW-HAs readily infiltrate tissues due to their viscoelasticity, which is highly dependent on the weight and concentration of HA, temperature, pH, and ionic strength. They can also undergo deformation while retaining their water and ion-binding abilities. HMW-HAs are naturally non-toxic, biocompatible, and biodegradable compounds [39]. In addition, they can be chemically modified on their hydroxyl or carboxylate functions to improve their mechanical properties by cross-linking or to reduce their degradation rate [40]. Thus, HMW-HAs appear as a good filler material. However, the beneficial roles of HMW-HAs seem to be abbreviated in pathological contexts associated with infection and/or inflammation that lead to the enzymatic and/or chemical fragmentation of HMW-HA into LMW-HA.

## 3. Pathological Context and LMW-HA

### 3.1. HA Degradation

Hyaluronidases (HYALs) are secreted by epithelial or stromal cells in the inflamed lung. These enzymes are involved in HMW-HA degradation into short HA fragments of 10 to 250 kDa (LMW-HA) [41]. In humans, six members belong to the HYAL family of endoglucosaminidases: HYAL1 to HYAL4, PH-20, and HYALP1 [42]. They are classified according to their pH-dependent activity: the acidic hyaluronidases HYAL1, 2, and 3 are active at pH = 3 to 4 [43,44,45,46], whereas PH-20 is a neutral hyaluronidase that is active at pH = 5 to 8 [47]. Their localization also differs, with HYAL1 and HYAL3 being lysosomal hyaluronidases, while HYAL2 and PH-20 are membrane-bound. PH-20 is a glycosylphosphatidylinositol (GPI)-anchored enzyme. HYALP1 is a non-functional hyaluronidase encoded by a pseudogene, and HYAL4 has recently been re-identified as a chondroitin sulfate (CS)-specific hydrolase that does not act on HA [48]. HYAL expression and activity are tightly regulated by pro-inflammatory cytokines, such as TNF-α and IL-1β [49]. Reciprocally, it has been shown that the use of HYAL inhibitors allows the downregulation of IL-18 production by allergen-stimulated keratinocytes [50], suggesting a relationship between the expression of inflammatory cytokines, HYAL, and LMW-HA. A decrease in the microenvironment pH due to the Na+/H+ exchanger has also been described as contributing to the activation of HYAL2, which is necessary for the fragmentation of CD44-bound HMW-HA [51]. The CD44–HMW-HA–HYAL2 complex promotes the cleavage of HMW-HA into 20 kDa fragments corresponding to about 50 disaccharide repeats. These HYAL2-generated fragments are internalized and found in endosomes. The complete digestion of HA fragments by β-exoglycosidases, β-glucuronidase, β-*N*-acetylglucosaminidase, and HYAL1 within lysosomal vesicles allows them to reach a minimum HA size of 200 Da [52]. Recently, the HYBID protein (HYaluronan (HA)-Binding protein Involved in HA Depolymerization/KIAA1199/CEMIP) has been identified as capable of degrading HMW-HA. The HYBID protein is a hyaladherin present in human dermal and arthritic synovial fibroblasts [53,54]. HMW-HAs are endocytosed, degraded in endosomes, and excreted into the extracellular space in the presence of HYBID and clathrin. This degradation is independent of HYAL2 and CD44 in skin fibroblasts [55]. HMW-HAs can also be degraded in a non-specific manner by reactive oxygen species (ROS) [56,57]. Electron paramagnetic resonance (EPR) spectroscopic studies support the hypothesis that ROS-mediated cleavage of HMW-HA may be due to the β-cleavage of radicals formed at the first carbon of one of the monosaccharides, the C3 of *N*-acetylglucosamine, or the C4 of glucuronic acid [56]. The degradation products are then recycled to reform a HA chain, which will give rise to LMW-HA secretion in a pathological context. Interestingly, it has been demonstrated that emphysematous lungs are characterized by a significant decrease in the percentage of HA. This result suggests that most of the inhaled deleterious factors promote HA degradation or downregulate HA turnover, resulting in the development of lung emphysema in aged lungs [58]. LMW-HAs are also detected in bronchoalveolar lavage (BAL) fluids from patients with persistent asthma associated with chronic inflammation, suggesting ECM-associated HA degradation [59].

### 3.2. Overall Impact of LMW-HA

HA polymers of 10 to 250 kDa are considered LMW-HA. Their effects differ from those of HMW-HA. They have been reported in many cell models, such as epithelial cells, smooth muscle cells, endothelial cells, or fibroblasts, when under inflammatory conditions, or cells associated with innate immunity. LMW-HAs can trigger an inflammatory response by acting as an endogenous danger signal [60]. The pro-inflammatory signaling associated with LMW-HAs is known to occur mainly through their interactions with Toll-Like Receptors (TLR2 and TLR4). TLR receptors can initiate different responses depending on the stimulus. For example, the TLR4/MD-2/CD44 complex recognizes LMW-HA in non-infectious inflammation [61], unlike the TLR4/MD-2/CD14 complex, which interacts with LPSs in infectious inflammation [62]. Moreover, LMW-HAs increase the expression of TLR4, MyD88, TRAF-6, iNOS, MMP-13, IL-1β, and IL-8, thus modulating both innate and acquired immunity [63]. LMW-HAs stimulate NF-κB and promote the expression of the pro-inflammatory cytokines IL-1β, TNF-α, IL-6, and IL-33 in synovial fibroblasts [64]. However, LMW-HA-associated effects appear to be tissue- or disease-dependent: the degradation of extracellular HMW-HA into LMW-HA following skin injury stimulates keratinocytes to release β-defensin-2 via TLR2 and TLR4 signaling involving a protein kinase C-dependent pathway mediated by c-Fos [65]. LMW-HA-induced activation of keratinocytes does not lead to an inflammatory response, as no IL-8, TNF-α, IL-1β, or IL-6 production was detected [65]. These examples illustrate the bioactive roles of LMW-HA, mainly in triggering a tissue-dependent response associated with a physiological or pathological context.

## 4. HMW-HA/LMW-HA Balance in the Lung 

Native HMW-HAs are hydrolyzed by HYAL1 and HYAL3 under physiological conditions [66] (Figure 1). HMW-HA intracellular degradation continues within lysosomal vesicles to reach a size of 200 Da. HAS-1 uses these degradation products to perform the neo-synthesis of HMW-Has, which are then secreted into the ECM. HMW-HAs play a protective role in epithelial cells and induce anti-inflammatory signaling by binding to CD44 [29]. HMW-HAs suppress macrophage M1 polarization, increase IL-10 production, and inhibit ROS-ASK1-p38/JNK-mediated epithelial apoptosis in fine particulate matter-induced lung inflammation and injury [29,30]. In contrast, ROS and HYAL, mainly HYAL2, fragment HMW-HAs into LMW-HAs, which are degraded into 200 Da fragments in lysosomes under pathological and inflammation-associated conditions. HAS3 then leads to the recycling and generation of HA fragments of 10 to 250 kDa, which mainly regulate pro-inflammatory activities. The microenvironment can alter the HMW-HA/LMW-HA balance. An increased expression of TNF-α and IL-1β, as well as the presence of ROS or an acidic pH, stimulate the production of LMW-HA [67]. The down-regulation of HAS1 expression or the upregulation of the expression of HYAL2 and HAS3 lead to the overproduction of LMW-HA. As an example, their production by fibroblasts is increased in the ventilated lung due to cell stretching, which promotes HAS3 expression and stimulates the tyrosine kinase signaling pathways, ultimately leading to increased expression of IL-8 [68]. This suggests that LMW-HA release could participate in the maintenance of the innate inflammatory response. The treatment of rats with bleomycin to induce alveolitis revealed a transient accumulation of LMW-HA in BAL, associated with an influx of T-cells, macrophages, and activated granulocytes into the lower airways [69,70,71]. Moreover, it was reported that HA, neutrophils, and IL-8 were more important in the bronchial sputum of COPD patients than in the sputum of non-COPD patients [72,73]. Finally, LMW-HAs have been shown to induce chemokine gene expression in alveolar macrophages in patients with idiopathic pulmonary fibrosis [74], whereas asthmatic macrophages showed a decrease in cell surface CD44 expression and an increase in TLR2, TLR4, and IL-8 expression [60]. Thus, LMW-HAs could be defined as potential therapeutic targets in the resolution of inflammatory processes.

## 5. LMW-HA and Lung Innate Immunity

The inflammation of the lung is a hallmark of respiratory diseases. Nonetheless, the molecular mechanisms associated with lung inflammation remain partially elucidated, in chronic obstructive lung disease (COPD), asthma, or infectious and/or genetic pathologies, such as bronchiectasis and cystic fibrosis (CF). In the human lung, the respiratory epithelium provides a physical barrier between the external environment and the underlying parenchyma. Its histology varies from the trachea to the alveoli, from a cylindrical pseudostratified epithelium composed of ciliated, basal, and secretory cells, to a monostratified epithelium containing type-1 and type-2 pneumocytes [75]. Lung inflammation is triggered by pathogens or by the inhalation of toxic particles or allergens. These stimuli favor the establishment of innate immunity, the first immune barrier.

The alarmins released by the damaged epithelium will alert immune cells via the increase in the epithelial expression of pro-inflammatory cytokines (TNF-α, IL-1β, and IL-8). These effectors will promote the recruitment of immune cells (neutrophils, dendritic cells (DC), and macrophages) to the inflammatory site. The involvement of LMW-HA in this phenomenon has been reported in numerous studies (Table 1). However, LMW-HA accumulation in the lung would maintain and/or amplify the inflammatory response, leading to the development of a deleterious inflammation. The responses associated with LMW-HA depend on the type of HA receptor that is stimulated. All LMW-HA–hyaladherin (TLR, CD44, HARE/Stab2, TSG-6, or RHAMM) interactions belong to the lung hyaluronasome.

### 5.1. Neutrophils

Neutrophils are leukocytes that play a central role in the immune response due to constant interactions with DC, macrophages, and lymphocytes. They synthesize and release numerous mediators, including pro-inflammatory cytokines (TNF-α, IL-1α, IL-1β, IL-12, Interferon-α (IFN-α), IFN-δ, and IL-6) and anti-inflammatory cytokines (IL-1 Receptor Antagonist, TGF-β). Neutrophils also produce growth factors (granulocyte colony-stimulating factor (G-CSF) and granulocyte-macrophage colony-stimulating factor (GM-CSF)), C-X-C chemokines (IL-8 and Interferon-γ inducible protein-10 (IP-10)), C-C chemokines (macrophage inflammatory protein 1 (MIP-1α and MIP-1β)), and fibrogenic and angiogenic factors (Vascular Endothelial Growth Factor (VEGF)) [95]. Neutrophils play also important roles in the innate immune response, in tissue repair, and the coordinated implementation of an optimal adaptive immune response [96]. They can detect chemical gradients of chemokines, such as IL-8, IFN-δ, and anaphylatoxin C5a, to direct their migration. IL-8 is the main chemoattractant involved in neutrophil migration and infiltration in the lung. IL-8 is secreted by fibroblasts, lung epithelial cells, and immune cells, including neutrophils, on which it has an autocrine action [95,97,98]. LMW-HAs increase the expression of IL-8 in vitro [68]. There were significant increases in IL-8, myeloperoxidase (MPO), and HA in the BAL fluid of chronic bronchitis patients [99]. Consequently, we can hypothesize that HA fragments enhance neutrophil chemotaxis through IL-8 production. The activity of neutrophils can be mediated by the HA–CD44 interaction. The contacts between the neutrophil membrane protein CD44 and the HA of the ECM limit neutrophil migration and infiltration [100]. CD44 expression may also be regulated under pathological conditions. The neutrophil expression of CD44 is significantly higher in the submucosa of COPD patients than in non-COPD smokers [101]. Since LMW-HAs play a pro-inflammatory role, one could assume their potential involvement in the regulation of CD44 expression by neutrophils, which, however, remains to be demonstrated. Nevertheless, the impact of LMW-HA on some functions of the neutrophils has been reported. LMW-HA treatment of mice inhibits the apoptosis of neutrophils via the PI3K/Akt1 pathway by upregulating Mcl-1 expression, which leads to triggering lung inflammation [78]. The microbicidal activity of neutrophils is achieved by the release of neutrophil elastase (NE), a serine protease with broad specificity, including elastin, fibronectin, laminins, collagens, and proteoglycans [102,103]. NE can accelerate the development of progressive obstructive airway disease in bronchiolitis obliterans due to its elastolytic activity [104], as well as the development of emphysema, as demonstrated in α1-antitrypsin-deficient mice [105,106]. Surprisingly, LMW-HAs may provide a protective role against the NE-mediated degradation of lung elastin in vivo independently of enzymatic inhibition [107,108]. The intratracheal administration of LMW-HA allows a significant reduction of NE-induced airway enlargement [76,109]. 

### 5.2. Macrophages

Monocytes and macrophages are immune cells that belong to the mononuclear phagocyte system. Circulating monocytes can migrate to inflamed tissues and differentiate into immature DC or macrophages under the influence of growth factors or cytokines. Monocytes serve as a lifelong reservoir, while macrophages derived from monocytes represent the immediate effectors of innate immunity. Alveolar and tissue macrophages provide an immediate defense against foreign agents and participate in the inflammation and the repair process of injured tissues by the selective release of cytokines that contribute to tissue remodeling. They also participate in the initiation and development of the adaptive immune response. The phenotypic changes of macrophages may depend on the molecular weight of HA, with a pro-inflammatory response for LMW-HA or a pro-resolving response for HMW-HA [81]. LMW-HAs promote macrophage activation by stimulating the production of pro-inflammatory cytokines and chemokines [74,80,82,85,90,110]. The NF-κB/IκB pathway plays a major role in this phenomenon and the propagation of the tissue inflammatory response [84]. HA fragments also induce IFN-β through a MyD88-independent pathway involving TLR4. It has been proposed that IFN-β expression is generated by LMW-HA/TLR4 stimulation through the intervention of TRIF, TBK1, and IRF-3 molecular intermediates, even in the absence of viral infection [86,110]. In addition to TLR4, TLR2/MyD88 signaling allows LMW-HA to activate the innate immune response by stimulating the expression of macrophage-associated cytokines MIP-1α in a NF-κB-dependent pathway via TRAF6, IRAK, and PKCζ [60]. This MIP-1α production, as well as that of MIP-1β and KC (IL-8), is, however, downregulated at both the mRNA and protein levels by IL-10 and IFN-δ in primary murine macrophages [79]. LMW-HA activation of macrophage CD44 can lead to an increased expression of inflammatory cytokines, such as TNF-α or IL-1β, as well as the production of growth factors, including macrophage-derived insulin growth factor (IGF-1) [74,111]. LMW-HAs are also involved in lipid metabolism, as they induce the phosphorylation of cytosolic phospholipase A2α (cPLA2α), ERK1/2, p38, and JNK via the TLR4/MyD88 pathways. This activation triggers an increase in the expression of COX2 and the production of PGE2, both playing an important role in immune and inflammatory phenomena [87]. Conversely, the HMW-HA treatment of LPS-stimulated macrophages decreases PGE2 and COX2 expression [87]. LMW-HAs regulate the expression of macrophage proteases by inducing plasminogen activator inhibitor-1 (PAI-1) and inhibiting urokinase activity in an inflammatory environment. This could explain the impaired fibrinolytic activity and the development of fibrosis in acute lung injury [83]. MMP-12 is a macrophage metalloelastase involved in tissue remodeling in COPD [112]. Its expression, secretion, and enzymatic activity are stimulated by LMW-HA in alveolar macrophages from bleomycin-treated rats [89]. Finally, LMW-HAs upregulate nitric-oxide synthase (iNOS) expression and iNOS activity through a pathway dependent on NF-κB, promoting the release of nitric oxide in murine alveolar macrophages of inflamed lungs [88]. However, no study has demonstrated that LMW-HAs either alter the bactericidal properties of macrophages or increase cellular cooperation between macrophages and lymphocytes. Altogether, LMW-HAs participate in the activation of lung macrophages and maintain their pro-inflammatory activities.

### 5.3. Dendritic Cells (DCs)

DCs are leukocytes specialized in the antigenic presentation to T lymphocytes. They are rare cells derived from the differentiation of monocytes. They are found in all tissues, but in greater quantities in the T-zones of lymphoid organs. DCs exist in an immature form and play the role of sentinels specialized in antigenic capture and the detection of danger signals (infection, inflammation, and necrosis). These signals induce DC maturation, as well as a considerable increase in the expression of major histocompatibility complex (MHC) class-II and costimulatory molecules. Mature DCs migrate to the T-zones of lymphoid organs and stimulate naive T-cells. Therefore, they play a central role in the control of the immune response. Studies on the impact of HA on DCs are sparse. However, it has been demonstrated that DC activation is increased by LMW-HA produced at sites of inflammation. HA fragments induce the immunophenotypic maturation of DC and increase their production of IL-1β, TNF-α, and IL-12 cytokines [91]. These effects are specific to HA fragments [91] and appear to depend on the TLR4 pathway [92]. Altogether, HA fragments may trigger the innate immune response at the DC level.

### 5.4. Lung Epithelial Cells

The respiratory epithelium actively participates in the innate immunity of the lung. Infection or epithelial damage leads to the loss of tissue homeostasis and causes the release of alarmins by the lung epithelial cells, thereby initiating an inflammatory response. One of the first epithelial responses is excessive mucus secretion to clear infection or foreign particles, including cigarette smoke components or fine particulate matter. LMW-HAs initiate pro-inflammatory signaling at the level of the respiratory epithelium through their interactions with hyaladherins. LMW-HAs can interact with the Stabiline-2 (Stab2)/HARE (Hyaluronic Acid Receptor for Endocytosis, half-length Stab2) receptor. Stab2 proteolysis leads to a 190 kDa receptor, HARE, which is involved in HA internalization thanks to the presence of endocytosis motifs [113,114,115]. Then, HARE-mediated HA endocytosis activates the NF-κB and ERK1/2 intracellular signaling pathways. LMW-HAs also have pro-inflammatory effects through the activation of TLR signaling [116]. TLR2 and TLR4 are notably involved in innate immunity in response to LPS in the respiratory epithelium. LMW-HAs activate these receptors, leading to the stimulation of signaling pathways associated with inflammation, i.e., NF-κB, MAPK, and JNK [93,117]. As a consequence, LMW-HAs enhance the expression and secretion of cytokines involved in the pulmonary inflammatory response, such as IL-8, IL-6, TGF-β2, and MMP-13 [61,94]. Interactions with other hyaladherins, such as CD44 and RHAMM, may explain LMW-HA effects on lung epithelial cells in many inflammatory diseases. The gel-forming glycoproteins MUC-5AC and MUC-5B mucins, conferring lubricating properties to mucus, are overexpressed by airway secretory cells in CF and COPD [118,119,120]. In this context, an upregulation of mucins expression by LMW-HA through the CD44 receptor has been reported. The LMW-HA–CD44 interaction promotes CD44–EGFR dimerization, MAPK activation, and MUC-5B upregulation [121]. LMW-HAs may also exert beneficial effects on mucociliary clearance. The fragments of HA improve the ciliary beat frequency (CBF) of tracheal epithelial cells [122] through their interactions with RHAMM, a microtubule-associated protein involved in cell motility, transformation, growth, and proliferation [122,123,124,125,126]. Moreover, the treatment of airway epithelial cells with LMW-HA increases the expression of the tight-junction protein *Zonula occludens*-1 (ZO-1), improves the transepithelial electric resistance of the epithelium [123], and can, therefore, participate in the maintenance of the epithelial barrier integrity. 

## 6. LMW-HA Signaling Inhibition 

The pro-inflammatory properties of LMW-HA could make it a potential therapeutic target for rapid return to a steady state. It is crucial to identify the molecular intermediates that inhibit LMW-HA signaling to counter their actions. For example, IL-1 Receptor-Associated Kinase-M (IRAK-M), mainly expressed by macrophages, DCs, and airway epithelial cells, could be an interesting target. This intracellular negative regulator of TLR signaling is involved in innate immune homeostasis [127]. Whether the IRAK-M regulation and downstream signaling pathways are different depending on TLR ligands (LMW-HA or LPS) remains to be elucidated. TSG-6 could also be a good candidate. This protein plays a key role in modulating the inflammatory response of bronchial epithelial cells and inflammatory cells. In particular, the binding of TSG-6 to CD44 through HA on inflammatory cells leads to the dissociation of CD44 from TLR2 and TLR4, ultimately leading to the inhibition of the TLR-dependent NF-κB signaling pathway [128,129,130]. It is of the utmost importance to use relevant in vitro experimental models demonstrating the deleterious effects of LMW-HA to assess the therapeutic potential of such candidates before testing the hypotheses in animal models receiving an airway instillation of HA fragments. The use of primary cells and their cultures in three-dimensional (3D) differentiated structures is crucial. The cultivation of human primary airway epithelial cells at the air–liquid interface (ALI) [131] or as spheroids [132] is the best option. These 3D culture methods allow epithelial cells to form junctional and polarized epithelia composed of specialized cells, such as basal, secretory, or ciliated cells. The resulting epithelia are the closest in vitro representation of in vivo functional airway epithelia exhibiting apical/basal-directed secretion (cytokines/chemokines, mucins, ions, growth factors,…) and coordinated beating cilia. The collection of cell secretions, total or fractionated cell proteins, or RNAs is easily achievable, giving the possibility of omics analysis. These 3D cultures also have the advantage of allowing the polarized co-culture of epithelial cells with other cell types, such as immune cells. The use of recombinant proteins/activators/inhibitors, and cutting-edge methods, such as CRISPR genome editing that allows the modulation of gene expression, will permit, in the future, testing the therapeutic benefits of IRAK-M or TSG-6 modulation to counter the pro-inflammatory action of a LMW-HA treatment in these in vitro models. 

The inhibition of HAS3, involved in the production of LMW-HA, could also be an adequate strategy to minimize the pro-inflammatory properties of HA fragments. Phosphodiesterase 3 inhibitor Milrinone decreases HAS3 expression and inhibits LPS-induced lung inflammation and lung injury in ventilated rats [133]. Furthermore, corticosteroids, such as betamethasone or budesonide, reduce the concentration of LMW-HA in BAL fluids by decreasing the expression of HAS, as well as the phosphorylation of ERK and Akt signaling in vivo [134]. 

Since hypoxia frequently goes together with lung inflammation, it may appear as an attractive target. HMW-HA is degraded by hyaluronidases, such as HYAL1, resulting in LMW-HA fragments detected in the blood of patients with obstructive sleep apnea [135]. In this context, the use of extracellular superoxide dismutase (SOD) as a treatment could be considered. The intratracheal injection of SOD in a mouse model of asbestos-induced pulmonary injury inhibits LMW-HA-mediated lung inflammation [136]. To validate the potential use of SOD as a treatment against LMW-HA-mediated inflammation, a model of 3D airway epithelial cultures exposed to a hypoxic environment (1.5% O_2_–5% CO_2_–94.5% N_2_) [137] or murine models exposed to 10% O_2_ [138] could be used, with or without concomitant treatment with LMW-HA. 

In addition to the deleterious pro-inflammatory functions of LMW-HA that could be interesting to counter, the beneficial activities of HA fragments may be enhanced. Interestingly, LMW-HAs play an anti-NE role [77]. This property could be leveraged in the treatment of diseases, such as COPD, to prevent elastin fragmentation and the development of emphysema [139,140].

## 7. Conclusions

HA is a major component of the ECM. It is synthesized into HMW-Has, which exhibits anti-inflammatory functions. Under pathological conditions, HMW-HAs are degraded to give rise to LMW-HAs, which show pro-inflammatory properties. LMW-HAs are involved in innate immunity by activating associated cells, including neutrophils, macrophages, DC, and respiratory epithelial cells, thereby participating in lung protection. However, infection or inflammation may amplify the action of LMW-HAs via interactions with hyaladherins, leading to an increase in the secretion of cytokines (TNF-α, IFN-β, IL-6), chemokines (IL-8), ROS, or enzymes (NE, MMP-9, MMP-10, MMP-12). Although LMW-HAs may promote lung hyper-inflammation, they also have beneficial impacts on the airway epithelium. By increasing the CBF, they improve the mucociliary clearance, and, by modulating intercellular junction protein expression, they strengthen the integrity of the epithelial barrier. LMW-HAs exhibit a dual function in the lung, through the maintenance/amplification of inflammation and the protection of the lung epithelium (Figure 2). Regardless, the presence of LMW-HAs in the lungs/bronchial secretions correlates with the onset of pulmonary diseases. The most recent example is the detection of LMW-HA in respiratory secretions collected from intubated patients with COVID-19 and cadaveric lung tissues from COVID-19 patients [141]. Therefore, LMW-HA must be considered as a marker of lung inflammation in respiratory diseases that could be targeted in future therapeutics. 

## Figures and Tables

**Figure 1 biomolecules-12-00658-f001:**
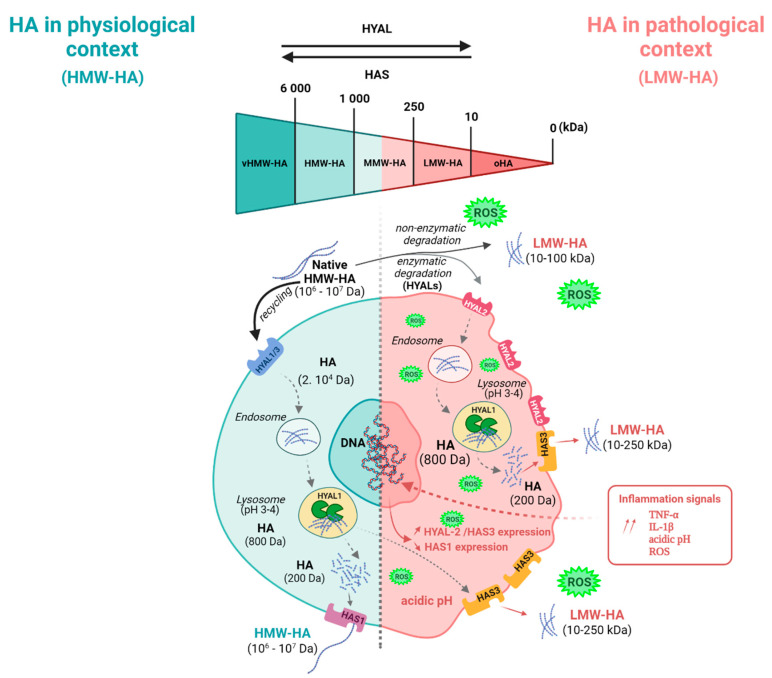
Catabolism and recycling of HA. In a physiological context, native HA (>1000 kDa) is mainly recycled by HYAL1 and HYAL3. Its intracellular degradation continues within lysosomal vesicles to reach a 200 Da size. HAS1 mainly performs the neo-synthesis of HMW-HA. In a pathological/inflammatory context, HMW-HAs are fragmented by ROS and HYAL (mainly HYAL2) to form LMW-HAs with a size between 10 and 250 kDa, then degraded into 200 Da-sized fragments in lysosomes. The neo-synthesis of LMW-HA (10–250 kDa) occurs through HAS3. Created with BioRender.com.

**Figure 2 biomolecules-12-00658-f002:**
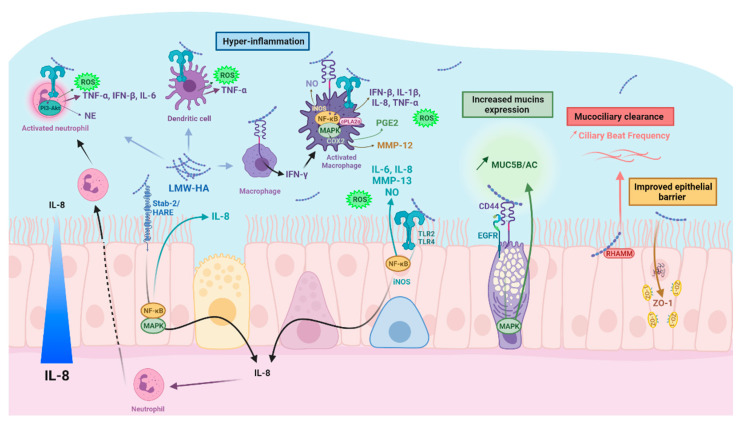
LMW-HA and lung hyaluronasome. LMW-HAs promote lung hyper-inflammation through hyaladherin interactions (CD44, TLR2, and TLR4, RHAMM, and HARE/Stab-2) and increase the secretion of cytokines (TNF-α, IFN-β, and IL-6), chemokines (IL-8), ROS, or enzymes (NE and MMPs). LMW-HAs also have beneficial impacts on the airway epithelium by increasing the CBF and by modulating intercellular junction protein expression. The dual functions of LMW-HAs maintain/amplify the lung inflammation while participating in the protection of the epithelium. Created with BioRender.com.

**Table 1 biomolecules-12-00658-t001:** LMW-HAs influence inflammatory marker expression.

Target Cell	LMW-HA (kDa)	Receptor	Pathways	Inflammatory Effector	References
Neutrophils					
	100–200	Not determined	Not determined	NE	[76,77]
	200	Not determined	PI3K/Akt1	IL-6	[78]
	200	Not determined	PI3K/Akt1	KC	[78]
	200	Not determined	PI3K/Akt1	MMP-9	[78]
	200	Not determined	PI3K/Akt1	MPO	[78]
	200	Not determined	PI3K/Akt1	Mcl-1	[78]
Macrophages					
	200	TLR2	NF-κB	MIP-1α/β	[60,79]
	80	TLR2 and TLR4	Not determined	IL-8	[74,80]
	100–150	TLR2 and TLR4	NF-κB	CXCL2	[80]
	5	TLR2 and TLR4	Not determined	TNF-α	[81,82]
	5–200	Not determined	NF-κB	NO	[81]
	Not determined	TLR2 and TLR4	Not determined	IL-1β	[81]
	60-200	Not determined	NF-κB	MMP-9	[82]
	Not determined	Not determined	Not determined	PAI-1	[83]
	Not determined	Not determined	Not determined	uPA	[83]
	80	TLR4	NF-κB	IFN-β	[84]
	100–150	CD44-TLR4	Not determined	IL-6	[85]
	100–150	CD44-TLR4	ERK1/2 and p38MAPK	MCP-1	[85]
	200	TLR4	IRF3	IFN-β	[86]
	200	cPLA2α Phosphorylation	ERK1/2, p38MAPK and JNK	COX2	[87]
	200	cPLA2α Phosphorylation	ERK1/2, p38MAPK and JNK	PGE2	[87]
	200	Not determined	NF-κB	iNOS	[88]
	200	Not determined	Not determined	MMP-12	[89]
	280	CD44	Not determined	IL-12	[90]
Dendritic Cells					
	80–200	TLR4	Not determined	IL-12	[91,92]
	80–200	TLR4	Not determined	TNF-α	[91,92]
	80–200	TLR4	Not determined	IL-1β	[91,92]
Lung Epithelial Cells					
	Not determined	TLR4	Not determined	IL-6	[61]
	Not determined	CD44/TLR4/MD2/ MyD88	NF-κB	MIP-2	[61]
	Not determined	CD44/TLR4/MD2/ Ras	NF-κB	MMP-13	[61]
	Not determined	CD44/TLR4/MD2/ Ras	NF-κB	TGF-β2	[61]
	200	Not determined	ERK1/2, AP-1, NF-κB and JNK	IL-8	[93,94]
	200	Not determined	NF-κB	IP-10	[93]

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
