# Peer review of "Lung Hyaluronasome: Involvement of Low Molecular Weight Ha (Lmw-Ha) in Innate Immunity"

_biomolecules, 2022, doi:10.3390/biom12050658_

Round 1

Reviewer 1 Report

The article add some more information on the biological function of Hyaluronic acid and it is acceptable for publication

Author Response

We thank reviewer 1 to judge our paper acceptable for publication in Biomolecules.

Reviewer 2 Report

Dear Authors,
your work gives an exhaustive overview of the role of HMW-HA and its fragmented product LMW.
The HA function has been an exponentially growing topic in recent years and many reviews have been produced on this. However, specific work on the role of LMW-IA in the lung has not been covered.
It is well written and organized work.
I have just a minor comment: although Figure 1 well schematizes the pro and anti-inflammatory functions of both high and low molecular weight HA, the anti-inflammatory effects of HMW-HA under physiological conditions should also be shown specifying the cell type and therefore the anti-inflammatory factors secreted by it and, regarding the pathological conditions, it should be specified which cells secrete the inflammatory factors shown in the figure.

Author Response

We thank reviewer 2 for these positive comments on our review.

We would like to draw his/her attention to the fact that Figure 1 intended to explain the catabolism and recycling of HA, under physiological and pathological conditions. The inflammatory factors, cell sources and cell targets are described in Figure 2.

Reviewer 3 Report

Here, Hoarau, Polette and Coraux presented a critical review discussing the involvement of low molecular weight hyaluronic acid (LMW-HA) in innate immunity. The concept behind the review is that HA, a major component of the extracellular matrix, is synthesised as a high molecular weight polymer (HMW-HA) which exerts anti-inflammatory function via binding to hyaladherins, maintaining the local homeostasis. Under pathological conditions (e.g. at inflammation and acidification conditions) neutral and acidic hyaluronidases secreted by the immune cells (mostly macrophages) degrade HMW-HA to LMW-HA possessing pro-inflammatory properties and closing the positive feedback loop. 

The paper comprehensively describes the beneficial role of HMW-HA and its therapeutic applications (e.g. aesthetic medicine, dermatology or orthopaedics), deleterious effects of LMW-HA through the TLR2 and TLR4 signaling cascades, response of various cell populations (neutrophils, macrophages, dendritic cells, and lung epithelial cells) to LMW-HA, and promising approaches to LMW-HA inhibition. 

The review is timely and written in a logical order. It can be accepted as is but the authors might also consider to add the methodological basis to investigate the pathophysiological role of LMW-HA, in particular in context of applying the pharmacological approaches to counteract its pathogenic effects. What are in vitro and in vivo models to investigate the detrimental effects of LMW-HA? Are cytokine (IL-6, IL-8, TNF-α, IFN-γ) and protease (NE, MMP-8/9/10/12) profiling by means of ELISA sufficient for the unbiased evaluation of cellular response to LMW-HA? What do we know from the -omics data obtained in the respective experiments (and have such experiments been conducted)? What are the most valuable and reliable biomarkers when assessing LMW-HA induced pulmonary dysfunction in vitro and in vivo? What is the influence of hypoxia, which frequently accompanies lung inflammation, on LMW-HA action? As hypoxia provokes further acidification which amplifies inflammation and HMW-HA degradation, it should play a negative synergistic role, potentiating the deleterious effects of LMW-HA. Which hypoxia models are the most useful to imitate this pathophysiological scenario?

Authors might discuss the methodological issues in the next section before the conclusions; again, the review is good and acceptable as is but these issues anyway are of major interest and can be targeted in this paper.

Author Response

We thank reviewer 3 for these positive comments on our review and  to judge our paper acceptable for publication in Biomolecules.

In order to ameliorate the manuscript according to the reviewer 3 comments, we have added a paragraph on models allowing to study the deleterious effects of LMW-HA and methods to counter them. We have also added some information on the impact of hypoxia on the mechanisms of action of LMW-HA. Finally, the English language has been carefully checked.